# Sesquiterpene Coumarins, Chromones, and Acetophenone Derivatives with Selective Cytotoxicities from the Roots of *Ferula caspica* M. Bieb. (Apiaceae)

**DOI:** 10.3390/ph17101254

**Published:** 2024-09-24

**Authors:** Fadıl Kaan Kuran, Gülsüm Altıparmak Ülbegi, Gülşah Gamze Arcan, Fatma Memnune Eruçar, Şule Nur Karavuş, Pınar Aksoy Sağırlı, Nur Tan, Mahmut Miski

**Affiliations:** 1Department of Pharmacognosy, Faculty of Pharmacy, İstanbul University, 34116 İstanbul, Türkiye; kaankuran@istanbul.edu.tr (F.K.K.); memnune.erucar@istanbul.edu.tr (F.M.E.); nurtan@istanbul.edu.tr (N.T.); 2Department of Pharmacognosy, Institute of Graduate Studies in Health Sciences, İstanbul University, 34116 İstanbul, Türkiye; 3Department of Biochemistry, Faculty of Pharmacy, İstanbul University, 34116 İstanbul, Türkiye; gulsum.altiparmakulbegi@istanbul.edu.tr (G.A.Ü.); gamze.arcan@istanbul.edu.tr (G.G.A.); aksoyp@istanbul.edu.tr (P.A.S.); 4Department of Pharmacognosy, School of Pharmacy, İstanbul Medipol University, 34810 İstanbul, Türkiye; sule.karavus@medipol.edu.tr

**Keywords:** *Ferula caspica*, sesquiterpenes, cytotoxicity, COLO 205, MCF-7, K-562, Bcl-xL, caspase-3/8/9, molecular docking

## Abstract

In search of selective cytotoxic compounds from *Ferula* species as potential leads for the treatment of various cancer diseases, a bioactivity-guided isolation study was performed on the roots of *Ferula caspica* M. Bieb. COLO 205 (colon), K-562 (leukemia), and MCF-7 (breast) cancer cell lines were used to monitor the cytotoxic activity of column fractions and determine the IC_50_ value of the active compounds. In addition to the seven known (**5**–**11**) compounds, four previously unknown compounds: kayserin A (**1**), kayserin B (**2**), 8′-*epi*-kayserin B angelate (**3**), and 3-*epi*-ferulin D (**4**) were isolated from the dichloromethane extract of the roots of *F. caspica*. Structure elucidation of the isolated compounds was carried out by extensive spectroscopic analyses such as 1D- and 2D-NMR spectroscopy, High-Resolution Mass Spectroscopy (HRMS), IR spectroscopy, and UV spectroscopy. Although all of the isolated compounds showed various degrees of cytotoxic activity on COLO 205, K-562, and MCF-7 cancer cell lines, the most potent compounds were identified in the following order: 1-Hydroxy-1-(1′-farnesyl)-4,6-dihydroxyacetophenone (HFDHAP, **11**), 3-*epi*-ferulin D (3EFD, **4**), and 7-desmethylferulin D (7DMFD, **6**). The cytotoxic activities of all three compounds were more potent than that of the reference compound cisplatin (Cis) against all tested cancer cell lines. Still, only HFDHAP (**11**) was more potent than the reference compound doxorubicin (Dox) against the MCF-7 cancer cell line. The mechanism of action of these three compounds was investigated on the COLO 205 cell line. The results indicated that compounds **4**, **6**, and **11** trigger caspase-3/8/9 activation and suppress the anti-apoptotic protein, Bcl-xL. Molecular docking studies confirmed the interactions of the three cytotoxic molecules with the active site of the Bcl-xL protein.

## 1. Introduction

Cancer, which arises from the uncontrolled proliferation of abnormal cells, is a complex array of diseases and the second most common cause of death in the United States. Breast, colon, and lymphoma cancers constitute the leading types of new cases and cancer-related deaths in the United States [1]. A major proportion of the cancer drugs approved by the FDA since 1981 were natural products or natural product-derived molecules [2]. Thus, exploring natural resources based on historical and traditional knowledge may lead to the discovery of novel anticancer drug substance candidates. Historically, the resin of various *Ferula* species from the Apiaceae family has been used to treat various cancer diseases [3,4].

Lately, our investigation of the root extracts of *Ferula huber-morathii* Peşmen from the subgenus *Dorematoides* (Rgl. et Schmalh.) Korovin of the genus *Ferula* L. afforded several sesquiterpene coumarin ethers with selective cytotoxic activity [5]. *Ferula caspica* M. Bieb. is another *Ferula* species from the subgenus *Dorematoides.* It is widely distributed in Western Mongolia, Western Siberia, Central Asian countries, the European section of Russia, Ukraine, the Caucasus region, and Türkiye’s Central and Eastern Anatolia regions [6,7]. Previously, three known sesquiterpene coumarins, umbelliprenin, farnesiferol A, and farnesiferol C, were reported in *F. caspica* [8,9]. Recently, three known sesquiterpene derivatives, a mixture of two phenylpropanoid esters, steroid mixtures, and flavonol glycosides, have been isolated from the aerial parts of *F. caspica* [10]. As part of our ongoing research on the bioactive constituents of *Ferula* species native to Anatolia [11,12,13,14,15,16,17,18,19,20,21], we investigated *F. caspica* for its cytotoxic compounds. Here, we report the isolation and structure elucidation of three previously undescribed novel sesquiterpene coumarin ethers, kayserin A (**1**), kayserin B (**2**), and 8′-*epi*-kayserin B angelate (**3**), as well as a novel sesquiterpene chromone called 3-*epi*-Ferulin D (**4**), along with seven known sesquiterpene compounds (**5–11**) (Figure 1).

## 2. Results and Discussion

### 2.1. Bioactivity-Directed Isolation of Cytotoxic Sesquiterpenes

The dichloromethane and methanol extracts of the roots of *Ferula caspica* were tested against COLO 205, K-562, and MCF-7 cancer cell lines, which showed that the cytotoxic compounds were in the dichloromethane extract; in contrast, the methanol extract of the root did not show any cytotoxic activity at a concentration up to 100 µg/mL (Table 1).

The dichloromethane extract of the roots of *Ferula caspica* was fractionated on a silica gel column (Appendix A). Column fractions were combined based on their cytotoxic activity and thin layer chromatography (TLC) profiles. Each combined fraction was further fractionated using various chromatographic techniques and their combinations (i.e., Sephadex LH-20 column chromatography, RP-18 flash chromatography, or prep. TLC). The following 11 cytotoxic sesquiterpene derivatives were isolated: kayserin A (**1**), kayserin B (**2**), 8′-*epi*-kayserin B angelate (**3**), 3-*epi*-ferulin D (**4**), ferulin D (**5**), 7-desmethylferulin D (**6**), colladonin (**7**), isosamarcandin (**8**), FFC (**9**), NDHAP (**10**), and HFDHAP (**11**).

### 2.2. Structure Determination of the Cytotoxic Sesquiterpene Derivatives

The structures of cytotoxic sesquiterpene derivatives of *Ferula caspica* were determined using extensive spectroscopic analysis techniques such as ^1^H-NMR, ^13^C-NMR, 2D ^1^H-^1^H correlation spectroscopy (COSY), nuclear Overhauser effect spectroscopy (NOESY), heteronuclear single-quantum coherence (HSQC), heteronuclear multi-bond connectivity (HMBC) spectroscopy (Table 2, Figure 2, Figure 3, Figure 4 and Figure 5, Appendix A). The identification of known compounds, ferulin D (**5**) [22], 7-desmethylferulin D (**6**) [23], colladonin (**7**) [24], isosamarcandin (**8**) [5], farnesyl furanocoumarin derivative (FFC) (**9**) [25], 1-(3′-nerolidyl)-4,6-dihydroxy acetophenone (NDHAP) (**10**) [26], and 1-hydroxy-1-(1′-farnesyl)-4,6-dihydroxy acetophenone (HFDHAP) (**11**) [26] (Figure 1), was accomplished by a comparison of their spectroscopic data with published data and by direct comparison with the reference compounds where available.

Kayserin A (**1**) was isolated as a white amorphous powder. The molecular formula of compound **1** was determined as C_25_H_33_O_5_ by the HRESIMS peak of the protonated molecule at *m*/*z* 413.2321 ([M + H]^+^, C_25_H_33_O_5_
^+^ cal. for 413.2328), indicating 10 degrees of unsaturation. The ^1^H- and ^13^C-NMR data of compound **1** (see Table 2) were similar to those of colladonin (**7**) [24], a well-known sesquiterpene coumarin ether previously isolated from several *Ferula* species [9], except for the signals of the coumarin section of compound **1**. In addition to the presence of a methyl signal due to an aromatic methoxy group at δ 3.86 ppm (s, 3H), along with two aromatic proton singlets at δ 3.84 and 3.85 ppm (each 1H, H-5, and H-8), the ^1^H-NMR spectrum of kayserin A (**1**) strongly suggests that the umbelliferon ether part of the colladonin (**7**) was replaced with scopoletin (i.e., 6-methoxylated umbelliferone) in compound **1**. Furthermore, the presence of a long-range interaction between the H-11′a and H-11′b proton signals and that of the C-7 carbon atom of scopoletin moiety in the 2D heteronuclear multi-bond connectivity (HMBC) spectrum (Figure 2 and Appendix A) confirms this structural difference between kayserin A (**1**) and colladonin (**7**). The proton interactions observed in the 2D nuclear. 

Overhauser effect spectroscopy (NOESY) spectrum (Figure 2 and Appendix A) of compound **1** and the almost identical chemical shifts of the carbon atoms constituting the drimane sesquiterpene parts of the colladonin (**7**) [24] and kayserin A (**1**) (see Table 2) strongly suggest the presence of identical stereochemistries between these compounds. Therefore, the structure of kayserin A should be as depicted **1** in Figure 1.

Kayserin B (**2**) was obtained as a white amorphous compound. The HRESIMS spectrum of compound **2** yielded a [M-H_2_O + H]^+^ peak at *m*/*z* 413.2311 (C_25_H_33_O_5_^+^ cal. for 413.2328), suggesting nine degrees of unsaturation for compound **2**. Similar to the structural relation observed between kayserin A (**1**) and colladonin (**5**), the ^1^H-NMR and ^13^C-NMR data of kayserin B (**2**) (Table 2) display a close resemblance to those of isosamarcandin (**8**) (Table 2, Appendix A). The presence of a methyl signal due to an aromatic methoxy group at δ 3.87 ppm (s, 3H) along with the partially overlapping of two aromatic proton singlets at δ 3.84 ppm (each 1H, H-5, and H-8) in the ^1^H-NMR spectrum of kayserin B (**2**) confirm that the umbelliferon ether part of the isosamarcandin (**8**) was replaced with scopoletin in compound **2**. In addition, the presence of a long-range interaction between one of the H-11′ proton signals and that of the C-7 carbon atom of the scopoletin moiety in the 2D heteronuclear multi-bond connectivity (HMBC) spectrum (Figure 3 and Appendix A) corroborates the presence of scopoletin instead of umbelliferone in kayserin B (**2**). The 2D NOESY correlations observed in compound **2** (Figure 3 and Appendix A) also confirms the same relative stereochemistry in the drimane sesquiterpenoid part of compound **2**. Thus, the structure of kayserin B (**2**) should be as shown in Figure 1.

8′-*epi*-Kayserin B angelate (**3**) was isolated as a white amorphous compound, and its molecular formula was determined as C_30_H_40_O_7_ via HRESIMS spectroscopy, as its sodium adduct molecular peak was found at *m*/*z* 535.2656 ([M + Na]^+^, calcd. for C_30_H_40_O_7_Na, 535.2672), suggesting eleven degrees of unsaturation for compound **3**. The ^1^H- and ^13^C-NMR spectra of compound **3** (Table 2) were similar to those of kayserin B (**2**) except for the presence of angelic acid-related ^1^H- and ^13^C-NMR signals (i.e., ^1^H-NMR: δ 6.05 ppm; qq: 1.4, 7.2; 1H: 1.89 ppm; quint: 1.4; 3H: 1.99 ppm; dq: 7.2, 1.4; 3H and ^13^C-NMR: δ 167.9, 128.4, 137.8, 15.9, and 20.8 ppm) in its ^1^H- and ^13^C-NMR spectra (Table 2), which suggest the presence of an angelic acid ester in compound **3**. The downfield shift of the C-3′ hydroxyl geminal proton (i.e., ca. 1.3 ppm) in the ^1^H-NMR of **3** strongly suggests that the angelic acid was esterified at the C-3′ hydroxyl group of compound **3**. Furthermore, the presence of a cross-correlation between the C-1″ carbonyl and H-3′ proton in the 2D HMBC spectrum of **3** (Figure 4 and Supplemental Material Appendix A) confirms the position of the angelic acid ester in compound **3** as C-3′. Another significant difference between compound **3** and kayserin B (**2**) was the chemical shift of the C-12′ methyl group in the ^13^C-NMR spectrum of compound **3**. Since the position of the angelic acid ester group was not close to the C-8′ atom of **3**, this difference was not due to the angelic acid ester at the C-3′ position. The 2D NOESY spectrum of compound **3** (Appendix A) clearly shows the presence of NOE correlations between the C-12′ and C-15′ methyl signals (Figure 4), which suggest a stereochemistry for the C-12′ methyl group in compound **3**. Hence, the structure of 8′-*epi*-kayserin B (**3**) should be as depicted in Figure 1.

3-*epi*-Ferulin D (**4**) was obtained as a white amorphous substance. Its molecular formula was identified as C_25_H_32_O_4_ by *m*/*z* 397.2374 in HRESIMS ([M + H]^+^, calcd. for C_25_H_33_O_4_, 397.2379). Overall, the ^1^H- and ^13^C-NMR spectra of compound **4** (Table 2) closely resemble those of ferulin D (**5**) and the C-3 epimer of 7-desmethylferulin D (**6**), farnesyl furanochromone derivatives isolated from *Ferula ferulioides* [22] and *Ferula mongolica* [23], respectively. The 2D NOESY spectrum of compound **4** (Figure 5 and Appendix A) corroborates the a stereochemistry for the C-3 methyl group via NOESY correlation between the C-3 methyl protons and C-1′ protons of the farnesyl side chain. Thus, the structure and stereochemistry of compound **4** (Figure 1) should be that of the C-3 epimer of ferulin D (**5**).

### 2.3. Cytotoxic Activities of the Sesquiterpene Derivatives Isolated from the Dichloromethane Extract of the Roots of Ferula caspica

The cytotoxic effects of eleven sesquiterpene derivatives isolated from the roots of *Ferula caspica* on COLO 205, MCF-7, and K-562 cancer cell lines were evaluated, and the results are given in Table 3 as the IC_50_ value.

Based on the results above, 1-hydroxy-1-(1′-farnesyl)-4,6-dihydroxyacetophenone (HFDHAP) (**11**) displayed the highest cytotoxic activity on the COLO 205 and MCF7 cell lines and was also the second-most active compound on the K-562 cell line. 3-*epi*-Ferulin D (**4**) was determined as the second-most active compound on COLO 205 and MCF-7, and the most active compound on the K-562 cell line was 7-desmethylferulin D (**6**). Compound **4** (C-3 epimer of **5**) showed higher activity than compound **5** in all cell lines. Kayserin A (**1**), kayserin B (**2**), and 8′-*epi*-kayserin B angelate (**3**) were newly discovered compounds and structurally similar to colladonin (**7**). Compound **1** exhibited approximately six times lower cytotoxic activity on COLO 205, three times lower cytotoxic activity on K-562, and two times lower cytotoxic activity on MCF-7 compared to compound **7**. Compound **2** had a good cytotoxic effect on COLO 205 compared to on other cell lines. However, it had a three-times-lower cytotoxic activity than compound **7**. Compound **3** showed a moderate cytotoxic effect against all cell lines. For COLO 205 and K-562, the cytotoxic effect of compound **3** was approximately six and two times weaker than that of compound **7**, respectively. The cytotoxic effect of compound **3** on MCF-7 cells was similar to that of colladonin (**7**). Of the compounds tested, isosamarcandin (**8**) exhibited the lowest cytotoxic activity in the COLO 205 and K-562 cell lines. Additionally, it demonstrated the second-lowest cytotoxicity in the MCF-7 cell line. Considering these findings, compounds **4** and **11** appear to be the most potent sesquiterpenes against cancer cell lines.

#### 2.3.1. Activities of 3-*epi*-Ferulin D (**4**), 7-Desmethylferulin D (**6**), and HFDHAP (**11**) on Caspase-3/8/9

The effects of compounds **4**, **6**, and **11** and cisplatin on caspase-3/8/9 activities were examined in the COLO 205 cell line. Concerning the results, the caspase-3/8/9 activities of compounds **4** and **6** were higher than those of the negative control in all concentrations. There was an increase in caspase-3 activities for all concentrations of compound **11**. However, caspase-8 and -9 activities were increased only at 130 µM and 26 µM concentrations. Furthermore, compound **4**, compound **6**, and compound **11** showed an increase in caspase-3 activity by 2.7-, 4.2-, and 2.9-fold; in caspase-8 activity by 2.2-, 2.2-, and 1.6-fold; and in caspase-9 activity by 2.6-, 2.7- and 1.8-fold, respectively, when compared with the negative control at the highest concentration (130 µM). Cisplatin (Cis) was a positive control for the caspase activities. It increased the caspase-3/8/9 activities by 6.73-, 1.8-, and 2.07-fold, respectively, compared with the control at the highest concentration (Figure 6).

#### 2.3.2. Western Blot Analysis

Bcl-xL is an anti-apoptotic protein and plays a role in the regulation of apoptosis by inhibiting the release of cytochrome c from mitochondria. Caspase-3 is an essential executioner for apoptosis processes in the cell due to its effect on disassembling the cells. Western Blot analysis revealed that compounds **4**, **6**, and **11** suppressed the level of Bcl-xL at all concentrations. The suppression was statistically significant (*p* < 0.05). Cisplatin (Cis), used as a positive control, showed statistical significance only at the highest concentration (*p* < 0.005). After treatment with different compounds, the procaspase-3 levels were significantly decreased only at the highest concentration of compound **4** and cisplatin in COLO 205 cells (*p* < 0.05). The rest of the data were not statistically significant (Figure 7).

One of the cytotoxic sesquiterpenes isolated from *F. caspica* was colladonin (**7**). It is a well-known cytotoxic sesquiterpene coumarin that has a cytotoxic effect on various cancer cells. We observed the following cytotoxic effects of colladonin (**7**) on COLO 205 (10.28 µM), K-562 (22.82 µM), and MCF-7 (43.69 µM) cancer cell lines. Our results revealed that colladonin (**7**) is more cytotoxic against the COLO 205 cell line than against the others. Some studies indicated that colladonin (**7**) has significant cytotoxic activity on the HCT116 (15.1 µM), HT-29 (13.3 µM), KM12 (2.5 µM), and COLO 205 (19 µM) colorectal cancer cell lines [27,28]. However, a slightly weaker cytotoxic effect of colladonin (**7**) on the HCT116 (47.4 µM) and COLO 205 (35.9 µM) cell lines was reported by Eruçar et al. [29], and a significantly weaker cytotoxic effect on the HCT116 (>100 µM) cell line was reported by Zhang et al. [30]. The cytotoxic effect of compound (**7**) was also identified on leukemia cell lines, such as the Jurkat (20 µM) [31] and K-562 (141.6 µM) [32] cell lines. Our result (K-562, 22.82 µM) supported that of the study of Shomirzoeva et al. (2021) [31], but it was not in agreement with that of Li et al. (2015) [32], considering that the same cell line (K-562) was used. Furthermore, the effect of compound **7** has a broad spectrum of activity on prostate, lung, renal, neuroectodermal, and glioblastoma cancer cells [28,29,33].

We have also investigated the cytotoxic effects of ferulin D (**5**), farnesyl furanocoumarin derivative (FFC, **9**), and 1-(3′-Nerolidyl)-4,6-dihydroxyacetophenone (NDHAP, **10**). The literature survey confirmed compounds **5**, **9**, and **10** cytotoxic activities on HepG2, MCF-7, and C6 cancer cell lines [22,34,35]. Meng et al. [22,34] determined that compound **10** had cytotoxic effects on the C6 (31 µM) and HepG2 (86.2 µM) cell lines but no cytotoxic effect on MCF-7 cells. Also, compound **9** showed a modest cytotoxic activity on C6 (20.1 µM), low cytotoxic activity on MCF-7 (60 µM), and no effect on HepG2; compound **5** did not affect any of these cell lines. In another study [35], compound **10** was significantly cytotoxic on MCF-7 cells (15.11 µM). We observed that compounds **5**, **9**, and **10** displayed moderate cytotoxic effects on COLO 205, K-562, and MCF-7 cells. Our test results of compound **9** agree with Jin et al.’s [35] results; however, the cytotoxic test results of compounds **5** and **10** do not agree with Meng et al.’s [22,34] results. Isosamarcandin (**8**) showed the weakest cytotoxic effect (209.2 µM) on colon cancer cells (COLO 205), and the results in the literature [27,30] are consistent with our observations (>50 µM).

### 2.4. Molecular Docking Studies

1-Hydroxy-1-(1′-farnesyl)-4,6-dihydroacetophenone (HFDHAP (**11**)), 7-desmethylferulin D (**6**), and 3-*epi*-ferulin D (**4**), the most active cytotoxic compounds of *Ferula caspica* (Table 3), were individually docked to the active site of Bcl-xL (PDB ID: 7LH7, 1.40 Å) [36] to examine their probable interactions with the amino acids in the binding site of the Bcl-xL. To verify the reliability of the docking study, the co-crystalized ligand (A1293102) of the Bcl-xL was redocked to the active site of a protein, and the root mean square deviation (RMSD) value was calculated. Depending on the ligand size, an RMSD value below 1.5 or 2 Å in size was regarded to have performed successfully [37]. RMSD was determined to be 0.561 Å, an acceptable value for docking model validation (Figure 8).

HFDHAP (**11**), 3-*epi*-ferulin D (**4**), and 7-desmethylferulin D (**6**) showed docking scores of −7.395, −5.696, and −5.141, respectively (Table 4). HFDHAP (**11**) and 7-desmethylferulin D (**6**) formed a hydrogen bond with ARG139 and π-π stacking interaction with PHE105. Two-dimensional and 3D representations of the docking positions of the compounds within the binding site of the Bcl-xL enzyme are given in Figure 9. Docking scores and interactions of the compounds and A1293102 within the amino acids of the active site of Bcl-xL are given in Table 4.

#### 2.4.1. Evaluation of the Molecular Docking Study Results

The binding site of Bcl-xL features two key pockets, P2 and P4. Our analysis of the interactions between the co-crystallized ligand A1293102 and the Bcl-xL enzyme’s active site determined that the benzothiazole moiety primarily occupies the P2 pocket. Specifically, the nitrogen atom of the benzothiazole ring engages in a hydrogen bond with LEU108, while the hydrogen atom of the amide group forms a hydrogen bond with SER106. These interactions are critical for the molecule’s binding affinity and selectivity. Additionally, the nitrogen atoms of the central thiazole ring establish a hydrogen bond with ARG139, and the carbonyl oxygen forms a hydrogen bond with ASN136. The remaining portion of the molecule effectively fills the P4 hydrophobic pocket, interacting with ALA93, PHE97, TYR101, and VAL141 through Van der Waals forces (Figure 10) [36]. Our docking studies corroborate these findings, reinforcing the literature’s evidence that A1293102 consistently forms these interactions within the P2 and P4 binding regions of the Bcl-xL enzyme. (Figure 10).

Based on the docking studies conducted with HFDHAP (**11**), 3-*epi*-ferulin D (**4**), and 7-desmethylferulin D (**6**), it was observed that HFDHAP (**11**) exhibited the most similar interactions to the natural ligand A1293102 and demonstrated superior docking results compared to the other two molecules. The inhibition region extends from the P2 pocket, where polar interactions occur, to the P4 pocket, characterized by hydrophobic interactions. An examination of A1293102′s interactions reveals that the hydrogen bonds formed between the benzothiazole group of the molecule and SER106 and LEU108 in the P2 pocket are critical for inhibition strength and selectivity. The inability of HFDHAP (**11**) and the other two molecules to interact with these amino acids is attributed to their insufficient size to occupy the region surrounding SER106 and LEU108. Another two significant interactions of the natural ligand involve hydrogen bonds with ASN136 and ARG139, which are also crucial for inhibition. HFDHAP (**11**) demonstrated this interaction with ARG139 via its carbonyl oxygen and phenolic hydroxyl group. Hydrophobic interactions are another important factor for Bcl-xL inhibition. As depicted in Figure 10, part of the natural ligand occupies the hydrophobic P4 pocket. HFDHAP (**11**) contains a linear farnesyl group attached via a carbonyl group. The farnesyl group is fully accommodated within the enzyme’s P4 pocket and forms Van der Waals interactions with ALA93, PHE97, and TYR101, similar to the natural ligand. Despite 3-*epi*-ferulin D (**4**) not forming any polar interactions, it effectively occupies the enzyme’s active site and establishes hydrophobic interactions. The interactions of 7-desmethylferulin D (**6**) resemble those of HFDHAP (**11**); it forms a single hydrogen bond with ARG139 via its carbonyl oxygen. Additionally, 7-Desmethylferulin D (**6**) exhibits a π-π stacking interaction with PHE105, similar to HFDHAP (**11**), but it does not occupy the hydrophobic P4 pocket as effectively as HFDHAP (**11**).

#### 2.4.2. ADME Studies

Given the crucial role of ADME (Absorption, Distribution, Metabolism, and Excretion) studies in enhancing the pharmacokinetic profiles of compounds during drug development, the pharmacokinetic attributes of the compounds employed in the molecular docking analysis were evaluated utilizing the QikProp module of the Schrödinger Molecular Modeling suite. Table 5 delineates the predicted ADME metrics for the compounds alongside their optimal value ranges [38,39]. All compounds exhibit molecular weight (MW) values of 372.503, 382.499, and 396.525, adhering to Lipinski’s rule of five as their MW is below 500, while the molecular weight of A1293102 is 972.116. The LogPo/w, a parameter indicating the solubility characteristics of compounds, should be within the range of −2 to 5. The LogPo/w values for the compounds and A1293102 were between 4.763 and 5.706, all within the acceptable range. The LogBB parameter, used to assess the ability of compounds to cross the blood–brain barrier, should lie between −3 and 1.2. The LogBB values for the compounds ranged from −1.897 to −0.257, meeting the specified acceptable range. PMDCK (Permeability Maden–Darby Canine Kidney) evaluates the apparent permeability of MDCK cells, expressed in nm/s, with higher values indicating better cell permeability. The PMDCK values for compounds ranged from 149.175 to 2365.400, which was considered favorable. The percentage of human oral absorption (%HOA) for all compounds is 100%, whereas the %HOA for A1293102 is 43.513%.

#### 2.4.3. Toxicological Evaluation

Toxic doses are generally expressed as LD_50_ values in mg/kg of body weight, where LD_50_ denotes the median lethal dose that results in the death of 50% of the test subjects following exposure to a substance. Toxicity categories are classified according to the Globally Harmonized System of Classification and Labeling of Chemicals (GHS) based on LD_50_ values:

Class I: Fatal if swallowed (LD_50_ ≤ 5 mg/kg)

Class II: Fatal if swallowed (5 < LD_50_ ≤ 50 mg/kg)

Class III: Toxic if swallowed (50 < LD_50_ ≤ 300 mg/kg)

Class IV: Harmful if swallowed (300 < LD_50_ ≤ 2000 mg/kg)

Class V: May be harmful if swallowed (2000 < LD_50_ ≤ 5000 mg/kg)

Class VI: Non-toxic (LD_50_ > 5000 mg/kg)

The potential toxicity of HFDHAP (**11**), 7-desmethylferulin D (**6**), and 3-*epi*-ferulin D (**4**), the most potent sesquiterpenoids of *Ferula caspica*, was evaluated based on their LD_50_ values using the ProTox-3.0 application [40] (Table 6). The compounds exhibit LD_50_ values ranging from 600 to 690 mg/kg, categorizing them as “Harmful if swallowed (Class IV: 300 < LD_50_ ≤ 2000 mg/kg)”. Consequently, higher doses of these three compounds are required to induce toxicity in healthy animals.

## 3. Materials and Methods

### 3.1. General Experimental Procedures

#### 3.1.1. Chemical Reagents, Solvents and Chromatographic Adsorbents

Anhydrous sodium sulfate (anhydr. Na_2_SO_4_), *p*-anisaldehyde, ethyl acetate (EtOAc), and dichloromethane (DCM) were purchased from Sigma-Aldrich (St. Louis, MO, USA). The hexane (Hxn), methanol (MeOH), ethanol (EtOH), acetonitrile (ACN), benzene, diethyl ether (Et_2_O), chloroform, cyclohexane, and sulfuric acid (H_2_SO_4_) were purchased from Merck (Darmstadt, Germany). The Milli Q ultrapure water (W) was obtained from Millipore (Billerica, MA, USA). Sephadex LH-20 was purchased from GE Healthcare (Chicago, IL, USA). Silica gel 60 (0.063–0.200 mm) for the column chromatography was purchased from Merck, as was the LiChroprep RP-18 (40–63 μm).

#### 3.1.2. Spectroscopic Analyses

Optical rotations were recorded using an Autopol V Plus polarimeter (Rudolph Research Analytical, Hackettstown, NJ, USA). Infrared spectra were acquired using an Alpha FT-IR spectrometer (Bruker, MA, USA). The absorbance spectra were obtained on a UV–Vis spectrophotometer UV-1700 PharmaSpec (Shimadzu, Columbia, MD, USA). NMR spectra were acquired on a Bruker BioSpin spectrometer (Rheinstetten, Germany) operating at 500 MHz for 1H and 125 MHz for 13C and equipped with a 5 mm probe in CDCl_3_. HRESIMS analyses for the compounds were measured using the Thermo Scientific Q Exactive LC-HRESIMS spectrometer (Thermo Scientific, Waltham, MA, USA).

#### 3.1.3. Column and Thin Layer Chromatography (TLC)

Supelco Silica gel 60 F254 PTLC plates (0.5, 1, and 2 mm; Merck, Darmstadt, Germany) were used for the preparative separations. Silica gel 60 F254 TLC plates (0.25 mm; Merck, Darmstadt, Germany) were used for the analytical separations. A UV lamp (Camag, Muttenz, Switzerland) with 254 and 366 nm wavelength detection capabilities was used to visualize the plates. Compounds without chromophore groups were detected by spraying plates with freshly prepared 10% *p*-anisaldehyde in 10% ethanolic sulfuric acid, followed by heating.

Sephadex LH-20 was purchased from GE Healthcare (Chicago, IL, USA). Silica gel 60 (0.063–0.200 mm) for column chromatography and LiChroprep RP-18 (40–63 µm) were purchased from Merck. Supelco Silica gel 60 F254 PTLC plates (0.5, 1, and 2 mm; Merck, Darmstadt, Germany) were used for the preparative separations.

### 3.2. Plant Material

Roots of *Ferula caspica* M. Bieb. were collected near the Hilmiye village in Kayseri province in June 2018. Prof. Emine Akalın identified the plant material, and a herbarium sample was deposited in the Istanbul University Herbarium (ISTE 116467).

### 3.3. Extraction and Isolation

Air-dried and coarsely pulverized root samples of *Ferula caspica* M. Bieb. (723.4 g) were successively extracted with dichloromethane and methanol. However, the plant material was initially subjected to a maceration (2 × 1 L for 1 h) at room temperature with the respective extraction solvent (i.e., dichloromethane and methanol) in a Soxhlet extractor before the initiation of continuous Soxhlet extraction.

The extracts obtained by maceration and continuous Soxhlet extraction were concentrated separately in vacuo in a rotary evaporator. Due to the close similarities between the TLC analyses of the extracts obtained by maceration vs. continuous Soxhlet extraction, they were combined and subjected to fractionation by column chromatography. The dichloromethane (DCM) extract was 56.1 g (yield 7.7%), and the methanol (MeOH) extract was 58.0 g (yield 8%).

The DCM extract (26.7 g) was subjected to a silica gel column (57 × 7 cm) with 0.04–0.5 mm particle size using an Hxn:EtOAc (*v*/*v* = 100:0 >> 0:100 with a 10% polarity increment) solvent system gradient to afford 22 fractions. The fractions were combined based on their TLC profiles.

Fraction 9 (3.77 g) was subjected to the Sephadex LH-20 column (70 × 2.5 cm) using an Hxn:DCM:MeOH (7:4:1 to 7:1:4 with 5% polarity increment) solvent system to afford 219 fractions. Compound **11** (HFDHAP, 2.4 mg) was obtained from subfractions 193–202. Compound **4** (3-*epi*-ferulin D, 1.8 mg) and compound **5** (ferulin D, 1.9 mg) were obtained from subfractions 22–25, and were then purified by preparative silica gel TLC (1 mm; Hxn:EtOAc, 7:3). Fractions 63–69 (326.4 mg) were subjected to a reverse-phase (RP-18) flash chromatography column (1.25 × 20 cm; W: MeOH, *v*/*v* = 50:50 >> 0:100) to yield 110 subfractions. Compound **10** (NDHAP, 2.3 mg) was obtained from subfractions 103–108. Compound **9** (farnesyl furanocoumarin derivative, 2.5 mg) was obtained from subfraction 94 and was then purified by preparative silica gel TLC (1 mm; Benzene:DCM:EtOAc:ACN, 8:8:2:1). Fractions 14–15 (1.53 g) were subjected to the Sephadex LH-20 column (100 × 2.5 cm) using an Hxn:DCM:MeOH (7:4:1 to 7:1:4 with 10% polarity increment) solvent system used to afford 62 fractions. The column chromatography (Sephadex LH-20, 120 × 2.5 cm) of fractions 34–47 (431 mg) (isocratic, Hxn:Chloroform:EtOH, 25:25:1) afforded 27 subfractions. Compound **6** (7-desmethylferulin D, 3.1 mg) was obtained from subfraction 23 (46.6 mg) and then was also purified by preparative silica gel TLC (2 mm, Hxn: EtOAc, 1:1). Compound **1** (kayserin A, 6.2 mg) and compound **7** (colladonin, 3.8 mg) were obtained from subfractions 1–2 (242 mg) and purified by RP-18 flash chromatography (1.25 × 20 cm column) using a W:MeOH gradient mobile phase (*v*/*v* = 50:50 >> 0:100). The column chromatography (Sephadex LH-20, 100 × 2.5 cm) of fraction 16 (299 mg) (gradient, Hxn:DCM: MeOH, 7:4.5:0.5) afforded 74 subfractions. Compound **8** (isosamarcandin, 1.8 mg) was isolated from subfraction 13. Subfraction 37 (9 mg) was also purified by preparative silica gel TLC (1 mm, Hxn: EtOAc, 6:4), and Compound **2** (kayserin B, 4.4 mg) was isolated. Fractions 16–54 (30 mg) were subjected to RP-18 flash chromatography (1.25 × 20 cm column) using a W:MeOH gradient mobile phase (*v*/*v* = 50:50 >> 0:100) to afford compound **3** (8′-*epi*-kayserin B angelate, 2.2 mg).

#### 3.3.1. Kayserin A (**1**)

Amorphous white powder; [α]D^21^: −93.75 (c, 1.67 mg/mL, CH_2_Cl_2_); UV (c, 0.015 mg/mL) (MeOH) λ_max_ (log ε) nm: 206 (4.70), 230 (sh) (4.33), 254 (sh) (3.86), 284 (sh) (3.75) nm, 296 (sh) (3.86), 344 (4.13), 370 (sh) (3.83); IR υ_max_ (NaCl) cm^−1^: 3468, 3080, 2938, 2869, 1720, 1644, 1614, 1560, 1514, 1465, 1424, 1386, 1278, 1248, 1198, 1170, 1146, 1096, 1031, 1007, 967, 928, 885, 821, 735, 700, 655, 590, 548, 497, 417 cm^−1^; ^1^H-NMR (500 MHz, CDCl_3_) (see Table 2), ^13^C-NMR (125 MHz, CDCl_3_): see Table 2, 2D-COSY, HSQC, HMBC, NOESY spectra (Appendix A), (+)-HRESIMS *m*/*z* 413.2321 [M + H]^+^ (calcd. for C_25_H_33_O_5_, 413.2328) (Appendix A).

#### 3.3.2. Kayserin B (**2**)

Amorphous white powder; [α]D^21^: +70.96 (c, 1.57 mg/mL, CH_2_Cl_2_); UV (c, 0.0016 mg/mL) (MeOH) λ_max_ (log ε) nm: 204 (5.32), 222 (sh) (4.82), 232 (sh) (4.67), 254 (sh) (4.37) nm, 298 (sh) (4.57), 326 (4.75), 348 (sh) (4.54); IR υ_max_ (NaCl) cm^−1^: 3382, 2917, 2849, 1725, 1611, 1577, 1539, 1512, 1465, 1383, 1353, 1278, 1232, 1198, 1126, 1061, 931, 835, 721, 635, 582, 526 cm^−1^; ^1^H-NMR (500 MHz, CDCl_3_) (see Table 2), ^13^C-NMR (125 MHz, CDCl_3_): see Table 2, 2D-COSY, HSQC, HMBC, NOESY spectra (Appendix A), (+)-HRESIMS *m*/*z* 413.2311 [M-H2O + H]^+^ (calcd. for C_25_H_33_O_5_, 413.2328) (Appendix A).

#### 3.3.3. 8′-*epi*-Kayserin B Angelate (**3**)

Amorphous white powder; [α]D^21^: −66.56 (c, 3.38 mg/mL, CH_2_Cl_2_); UV (c, 0.007 mg/mL) (MeOH) λ_max_ (log ε) nm: 203 (4.00), 221 (sh) (3.41), 290 (sh) (3.34), 317 (3.66) nm, 342 (sh) (3.22), 366 (3.16), 385 (sh) (2.94); IR υ_max_ (NaCl) cm^−1^: 3506, 2917, 2849, 1725, 1611, 1577, 1539, 1512, 1465, 1383, 1353, 1278, 1232, 1198, 1126, 1061, 931, 835, 721, 635, 582, 526 cm^−1^; ^1^H-NMR (500 MHz, CDCl_3_) (see Table 2), ^13^C-NMR (125 MHz, CDCl3): see Table 2, 2D-COSY, HSQC, HMBC, NOESY spectra (Appendix A), (+)-HRESIMS *m*/*z* 535.2656 [M + Na]^+^ (calcd. for C_30_H_40_O_7_Na, 535.2672) (Appendix A).

#### 3.3.4. 3-*epi*-Ferulin D (**4**)

Amorphous white powder; [α]D^21^: −35.39 (c, 1.13 mg/mL, CH_2_Cl_2_); UV (c, 0.005 mg/mL) (MeOH) λ_max_ (log ε) nm: 203 (3.93), 208 (sh) (3.87), 242 (sh) (3.44), 250 (sh) (3.39) nm, 294 (3.64), 308 (sh) (3.52), 351 (3.26), 385 (sh) (2.94); IR υ_max_ (NaCl) cm^−1^: 2918, 2850, 1735, 1616, 1555, 1461, 1376, 1245, 1152, 1093, 1028, 839, 775, 718, 464 cm^−1^; ^1^H-NMR (500 MHz, CDCl_3_) (see Table 2), ^13^C-NMR (125 MHz, CDCl_3_): see Table 2, 2D-COSY, HSQC, HMBC, NOESY spectra (Appendix A), (+)-HRESIMS *m*/*z* 397.2374 [M + H]^+^ (calcd. for C_25_H_33_O_4_, 397.2379) (Appendix A).

### 3.4. Cell Culture Conditions

Human colorectal cancer, breast cancer, and chronic myelogenous leukemia cell lines [COLO 205 (CCL-222), MCF-7 (HTB-22), and K-562 (CCL-243)] were obtained from the American Type Culture Collection. The COLO 205, MCF-7, and K-562 cells were maintained in Roswell Park Memorial Institute (RPMI 164, Wisent, Montreal, QC, Canada), Eagle’s Minimum Essential Medium (EMEM, Wisent, Montreal, QC, Canada), and Iscove’s Modified Dulbecco’s Medium (IMDM, Wisent, Montreal, QC, Canada) supplemented with 10% fetal bovine serum (Capricorn, Ebsdorfergrund, Germany), 100 U/mL penicillin, and 100 µg/mL streptomycin (Wisent, Montreal, QC, Canada), respectively. All cell lines were cultured at 37 °C in a humidified atmosphere with 5% CO_2_.

### 3.5. Cytotoxic Activity (MTS Assay)

The cytotoxic activities of the compounds isolated from *F. caspica* were carried out by the MTS Assay (Promega, Madison, WI, USA). The cells seeded in 96-well plates (COLO 205, 3 × 104; MCF-7 5 × 104 and K-562, 3 × 104) were exposed to different concentrations of the compounds and incubated for 72 h. Cisplatin (Cis) (sc-200896, Santa Cruz Biotechnology, CA, USA) and doxorubicin HCl (Dox) (Sigma, St. Louis, MO, USA) were used as a positive control. At the end of the incubation, MTS/PMS (Sigma, St. Louis, MO, USA) mixture was added to each well. After 1–4 h, the absorbance was read on a microplate reader (Biotek, Winooski, VT, USA) at 490 nm. The cytotoxic activities of the compounds were expressed as IC_50_ value. The inhibition % of cell proliferation was calculated using the formula: Inhibition % = [1 − (A490test/A490cont)] × 100, where A490test = absorbance of the test sample and A490cont = absorbance of the control sample [41]. The values of the 50% inhibition of the cell proliferation (IC_50_) were calculated by locating the *x*-axis values corresponding to one-half of the absorbance values of the sample.

#### 3.5.1. Caspase Activities

The caspase-3/8/9 activities were ascertained by following the protocol of caspase-3/8/9 colorimetric assay kits (Biovision, K106, K113, K119, Mountain View, CA, USA). Briefly, the cells treated with the compounds for 24 h were harvested and lysed with RIPA (50 mM Tris-HCl, pH 8.0, 150 mM NaCl, 1% Nonidet P-40, 0.5% sodium deoxycholate, and 0.1% SDS) cell lysis buffer. The lysed cells were centrifuged at 22,000× *g* for 30 min, and the supernatant was collected in a tube. The samples (200 µg protein) were incubated with caspase substrates (final concentration, 200 µM) at 37 °C for 2 h and read on a microplate reader at 405 nm. The fold-increase in enzyme activity was detected by comparison with the untreated control.

#### 3.5.2. Western Blot Analysis

To appraise the effect of the compounds on procaspase-3 and bcl-xL protein levels, COLO-205 cell lines were incubated with the compounds for 24 h. At the end of the incubation period, the treated cells were harvested. After that, the harvested cells were washed with cold PBS and lysed in RIPA buffer. The samples were centrifuged at 24,000× *g* for 45 min at 4 °C, and the supernatants were used to determine procaspase-3, Bcl-xL, and β-actin protein levels by Western blot analysis. The cellular proteins were separated on sodium dodecyl sulfate–polyacrylamide gels (SDS-PAGE, AnyKD mini protean TGX precast protein gel, Bio-Rad, Hercules, CA, USA, 4569033). The separated proteins were transferred to the Polyvinylidene difluoride (PVDF) membrane using the Trans-Blot Turbo™ transfer system (Bio-Rad, Hercules, CA, USA). The membrane was then blocked with a 5% non-fat milk-blocking buffer and consecutively incubated with rabbit monoclonal Bcl-xL antibodies (1:20,000, ab17884, Abcam, Cambridge, MA, USA), rabbit monoclonal caspase-3 antibodies (1:3000, ab3235, Abcam, Cambridge, MA, USA), and rabbit monoclonal beta-actin antibodies (1:10,000, M01263, BOSTER, Beijing, China) overnight at 4 °C. Furthermore, the membrane was washed with TBST solution and incubated with anti-rabbit IgG-HRP-conjugated (1:5000; sc-2357, Santa Cruz Biotechnology, CA, USA, and BA1054, BOSTER, Beijing, China) antibodies for 1 h at room temperature. Finally, the membrane was incubated with a chemiluminescent substrate for 5 min, and the protein bands were visualized by an imaging system (Vilber Lourmat, Fusion FX5, Marne-la-Vallée, France).

#### 3.5.3. Statistical Analysis

Statistical data analysis was performed using an Independent Sample *t*-test, with a *p*-value < 0.05 considered statistically significant. Each experiment was performed three times.

### 3.6. Molecular Docking

The crystal structure of the Bcl-xL protein in complex with A1293102 (PDB ID: 7LH7, resolution: 1.40 Å) [36] was accessed from the RCSB Protein Data Bank. The enzyme’s structure was prepped for docking analyses using the multistep Protein Preparation Wizard in the Schrödinger Software Suite [42]. Optimization of the protein structure involved adding missing hydrogen atoms and removing water molecules, heteroatoms, and co-factors, except for native ligands. The protein’s ligand binding site was identified by constructing a receptor grid box based on the atomic coordinates of the native ligand using the Receptor Grid Generation tool in Glide (Schrödinger Release 2022-3: Glide, Schrödinger, LLC, New York, NY, USA, 2022) [43,44,45]. The compounds were then docked into the binding site using Glide, with the docking process conducted in Standard Precision (SP) mode. The compounds were sketched using the 2D Sketcher tool in Schrödinger Maestro. To generate energy-minimized conformations and tautomers at pH 7.0 ± 2.0, the LigPrep module of the Schrödinger Suite (Schrödinger Release 2022-3: LigPrep, Schrödinger, LLC, New York, NY, USA, 2022) was utilized, employing Epik for the process. All resulting conformations and tautomers were subsequently minimized using the OPLS3 force field.

### 3.7. In Silico ADME Studies

The pharmacokinetic properties of the sesquiterpene coumarins were predicted using the QikProp module from the Schrödinger Software Suite (QikProp, Schrödinger, LLC, New York, NY, USA, 2022).

### 3.8. Toxicology Studies

The application of ProTox-3.0 was assessed to predict the toxicity of HFDHAP (**11**), 7-desmethylferulin D (**6**), and 3-*epi*-ferulin D (**4**) [40]. ProTox-3.0 predicts various toxicological endpoints by employing a combination of molecular similarity, fragment propensity, common characteristics, and a machine learning strategy termed fragment similarity-based CLUSTER cross-validation across 61 predictive models. These endpoints encompass acute toxicity, organ-specific toxicity, toxicological impacts, molecular initiating events, metabolic pathways, adverse outcome pathways (Tox21), and toxicity targets.

## 4. Conclusions

A bioactivity-directed isolation study led to the isolation of eleven cytotoxic sesquiterpenoid compounds from the dichloromethane extract of the roots of *Ferula caspica.* The structures of these compounds were determined using extensive spectroscopic techniques such as 1D- and 2D-NMR, HRESIMS, IR, and UV spectroscopic analyses. In addition to the seven known sesquiterpenoid compounds (**5–11**), four previously undescribed cytotoxic sesquiterpenes, kayserin A (**1**), kayserin B (**2**), 8′-*epi*-kayserin B angelate (**3**), and 3-*epi*-ferulin D (**4**), were described. Unlike previously described sesquiterpene coumarin ethers from *Ferula* species that are the derivatives of umbelliferone, kayserin A (**1**), kayserin B (**2**), and 8′-*epi*-kayserin B angelate (**3**) all share scopoletin moiety as the coumarin portion of these compounds. Scopoletin containing drimane sesquiterpene coumarin ether derivatives are known in the Asteraceae family [46], but so far, they have not been described in the Apiaceae family. Compounds **1–3** were the first examples of this type of compound described in the *Ferula* species and the Apiaceae family. An acyclic sesquiterpene coumarin ether, farnesylscopoletin, was recently described from an Apiacean plant, *Heptaptera triquetra* [47]. 

This study is the first report of compounds **1**, **2**, **3**, **4**, **6**, and **11**’s cytotoxic activities against the COLO 205, MCF-7, and K-562 cancer cell lines. Furthermore, the cytotoxic activity mechanism of compounds **4**, **6**, and **11** was investigated, which revealed that these compounds exert their cytotoxic activity by triggering caspase activity and suppressing the anti-apoptotic protein, Bcl-xL. The molecular docking studies also corroborated these observations.

## Figures and Tables

**Figure 1 pharmaceuticals-17-01254-f001:**
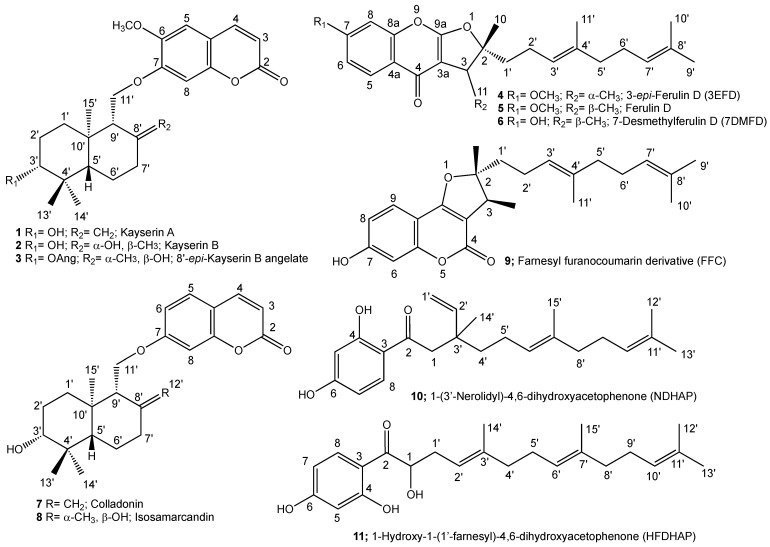
Structures of the cytotoxic sesquiterpenoids of *Ferula caspica*.

**Figure 2 pharmaceuticals-17-01254-f002:**
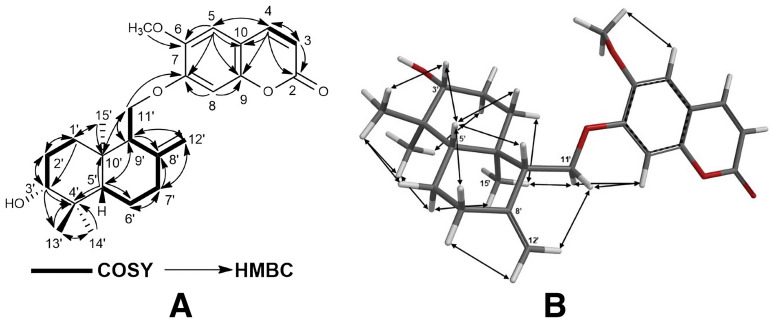
^1^H-^1^H-COSY and ^1^H-^13^C HMBC (**A**), and NOESY correlations (**B**) of kayserin A (**1**).

**Figure 3 pharmaceuticals-17-01254-f003:**
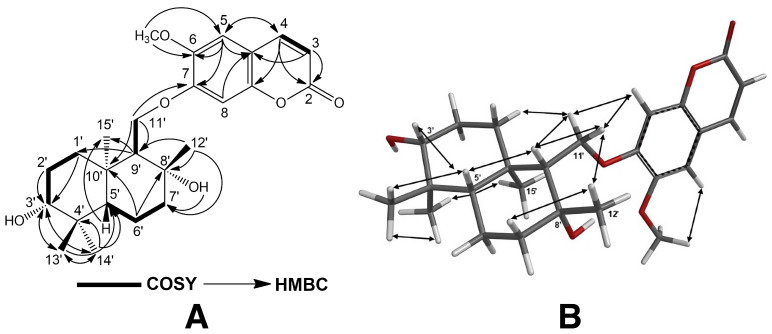
^1^H-^1^H-COSY and ^1^H-^13^C HMBC (**A**), and NOESY correlations (**B**) of kayserin B (**2**).

**Figure 4 pharmaceuticals-17-01254-f004:**
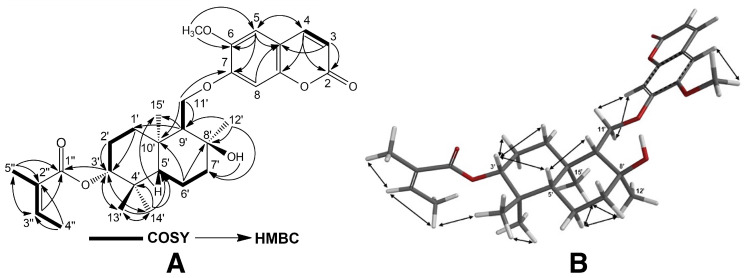
^1^H-^1^H-COSY and ^1^H-^13^C HMBC (**A**), and NOESY correlations (**B**) of 8′-*epi*-kayserin B angelate (**3**).

**Figure 5 pharmaceuticals-17-01254-f005:**
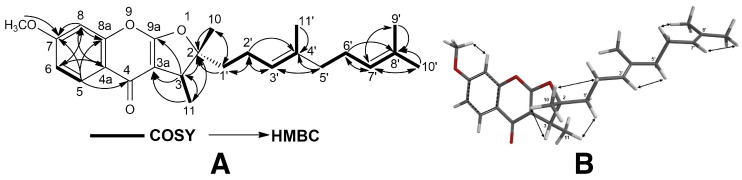
^1^H-^1^H-COSY and ^1^H-^13^C HMBC (**A**), and NOESY correlations (**B**) of 3-*epi*-ferulin D (**4**).

**Figure 6 pharmaceuticals-17-01254-f006:**
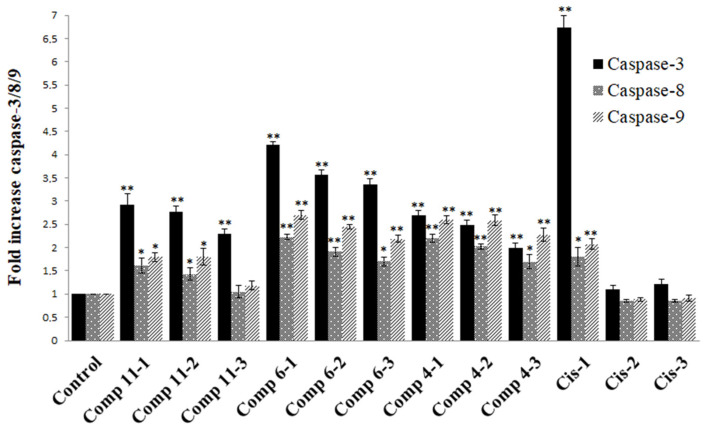
The effect of compounds and cisplatin on caspase-3/8/9 activations in COLO 205 cell lines. * *p* < 0.05; ** *p* < 0.005. Cis: cisplatin; Comp: 1 = 130 µM, 2 = 26 µM, 3 = 2.6 µM.

**Figure 7 pharmaceuticals-17-01254-f007:**
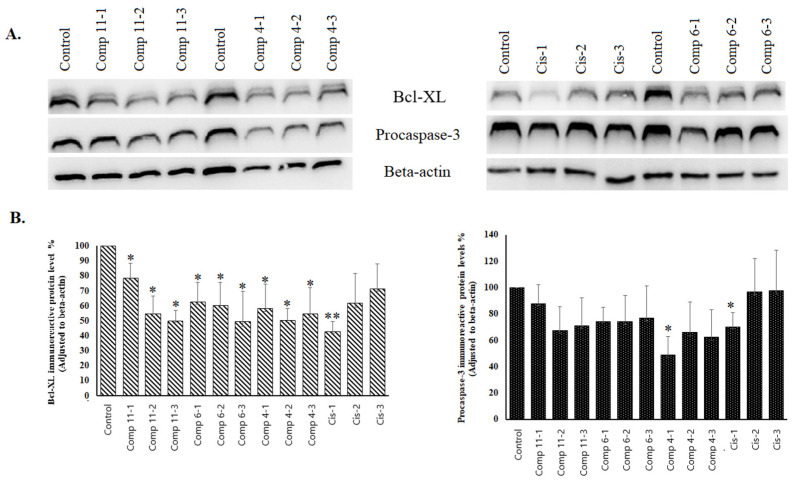
The effect of the compounds on Bcl-xL and procaspase-3 in immunoreactive protein levels in COLO 205 cell lines. (**A**) Bcl-xL, procaspase-3, and β-actin immunoreactive protein bands; (**B**) Bcl-xL and procaspase-3 immunoreactive protein levels in COLO 205 cell lines. * *p* < 0.05; ** *p* < 0.005. Cis: cisplatin; Comp: compound 1 = 130 µM, 2 = 26 µM, 3 = 2.6 µM.

**Figure 8 pharmaceuticals-17-01254-f008:**
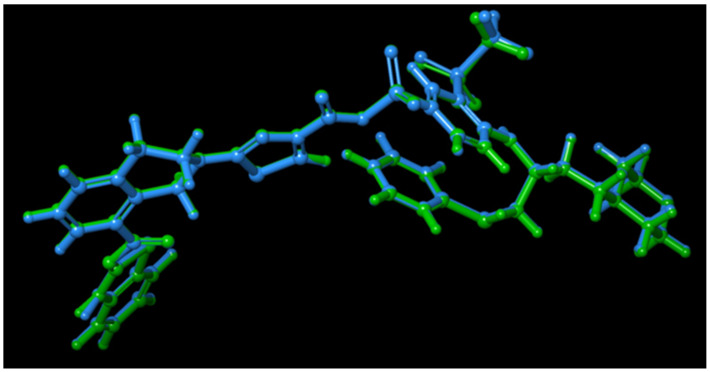
Validation of the docking models: superposition of docked pose (green) and experimental binding conformation (blue) of A1293102 at the binding site of Bcl-xL.

**Figure 9 pharmaceuticals-17-01254-f009:**
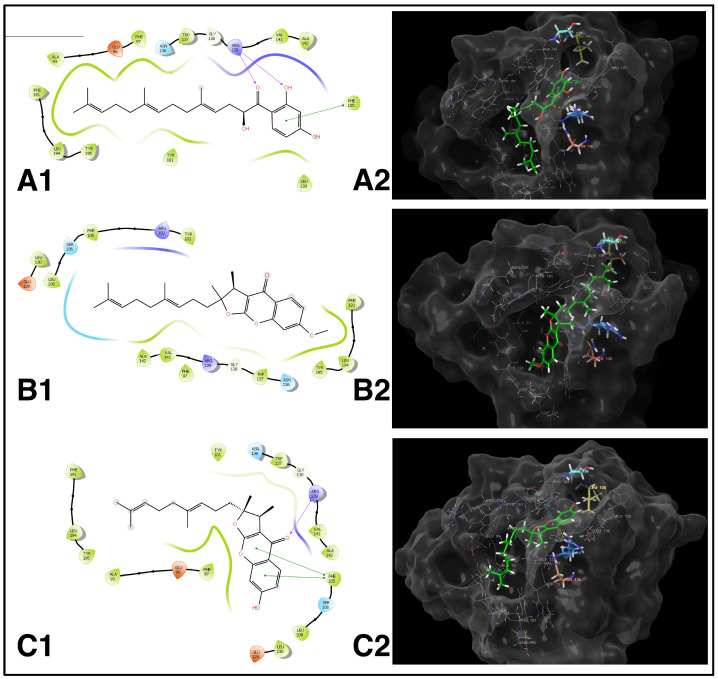
(**A1**,**A2**): HFDHAP (**11**) and its 2D–3D interactions with the active site of Bcl-xL. (**B1**,**B2**): 3-*epi*-Ferulin D (**4**) and its 2D–3D interactions with the active site of Bcl-xL. (**C1**,**C2**): 7-Desmethylferulin D (**6**) and its 2D–3D interactions with the active site of Bcl-xL.

**Figure 10 pharmaceuticals-17-01254-f010:**
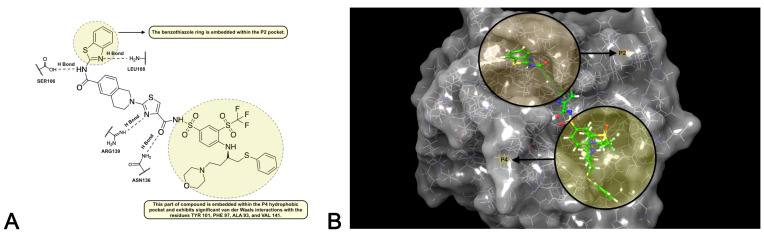
(**A**): The binding pockets P2 and P4 and the interactions of A1293102 in 2D. (**B**): The binding pockets P2 and P4 and the interactions of A1293102 in 3D.

**Table 1 pharmaceuticals-17-01254-t001:** Cytotoxic activities of *Ferula caspica* root extracts.

Extracts	IC_50_ (µg/mL) *		
	COLO 205	K-562	MCF-7
Dichloromethane Extract	23.64 ± 1.06	25.38 ± 1.64	35.97 ± 1.45
Methanol Extract	>100	>100	>100
Cisplatin (Cis) **	12.21 ± 0.34	9.73 ± 0.30	35.52 ± 0.76

* The cytotoxic effect of the extracts and cisplatin (Cis) were determined for 72 h. ** Positive control.

**Table 2 pharmaceuticals-17-01254-t002:** ^1^H NMR (500 MHz) and ^13^C NMR (125 MHz) data of compounds **1**, **2, 3,** and **4** (in CDCl_3_, δ in ppm, *J* in Hz).

Position	Kayserin A (1)		Kayserin B (2)		8′-*epi*-Kayserin B Angelate (3)		3-*epi*-Ferulin D	
	^1^H NMR	^13^C NMR	^1^H NMR	^13^C NMR	^1^H NMR	^13^C NMR	^1^H NMR	^13^C NMR
2	-	161.6	-	161.5	-	161.5	-	95.6
3	6.27; d; 9.4; 1H	113.5	6.29; d; 9.5; 1H	113.9	6.29; d; 9.5; 1H	113.9	3.29; q; 7; 1H	43.4
3a	-	-	-	-	-	-	-	99.3
4	7.60; d; 9.4; 1H	143.4	7.61; d; 9.5; 1H	143.4	7.62; d; 9.5; 1H	143.4	-	175.5
4a	-	-	-	-	-	-	-	118.0
5	6.84; s; 1H	109.0	6.84; s; 1H *	108.0	6.84; s; 1H	107.90	8.11; d; 8.8; 1H	127.0
6	-	147.0	-	146.7	-	146.4	6.96; dd; 2.5, 8.8; 1H	113.4
7	-	152.7	-	151.7	-	151.6	-	163.1
8	6.85; s; 1H	101.1	6.84; s; 1H *	100.7	6.91; s; 1H	100.81	6.83; d; 2.5; 1H	101.1
8a	-	-	-	-	-	-	-	153.3
9	-	150.20	-	150.0	-	150.0	-	-
9a	-	-	-	-	-	-	-	166.4
10	-	111.50	-	111.9	-	111.9	1.5; s; 3H	25.5
11							1.33; d; 7; 3H	14.2
-OCH_3_	3.86; s; 3H	56.9	3.87; s; 3H	56.5	3.88; s; 3H	56.4	3.88; s; 3H	55.9
1′α	1.82; dt; 3.4,13.1; 1H	37.2	1.86; m; 1H **	38.3	1.82; m; 1 H *	38.2	1.91; ddd; 5.8, 10.7, 14.3; 1H	35.2
1′β	1.48; m; 1H *		1.12; m; 1H		1.42; m; 1H **		1.67; m; 1H *	
2′α	1.62; ddd; 1.7, 3.4; 12.3; 1H	27.9	1.69; m; 2H ***	27.2	1.69; m; 1H ***	23.8	2.20; m; 2H	22.8
2′β	1.73; dq; 13; 3.8; 1H **				1.82; m; 1H *			
3′	3.30; dd; 4.3, 11.6; 1H	78.7	3.24; dd; 5.8, 10.5; 1H	78.9	4.60; dd; 4.2, 11.7; 1H	80.0	5.09; tquint; 6.7, 1.4; 1H	124.3
4′	-	39.3	-	39.1	-	38.05	-	136.2
5′	1.18; dd; 2.8, 12.6; 1H	54.4	0.88; dd; 2.1, 12.3; 1H	55.2	1.15; dd; 2.1, 12.1; 1H	55.05	2.00; m; 2H	39.8
6′α	1.45; m; 1H *	23.6	1.82; m; 1H **	18.4	1.39; qd; 3.4, 11.4; 1H **	19.7	2.08; m; 2H	26.8
6′β	1.78; dq; 10.5, 2.7; 1H **		1.58; dq; 2.5, 13.9; 1H		1.71; m; 1H ***			
7′α	2.45; ddd; 2.4, 4.4, 13.3; 1H	37.6	1.86; m; 1H **	42.6	1.97; m; 1H ****	43.3	5.15; brtq; 7.1, 1.1; 1H	123.4
7′β	2.11; td; 13.2, 5.2; 1H		1.44; td; 13.5, 3.7		1.64; td; 3.9, 13.2; 1H			
8′	-	146.42	-	72.5	-	72.4	-	131.6
9′	2.31; brt; 5.7; 1H	54.6	1.33; brdd; 1.8, 4.5; 1H	57.0	1.97; m; 1H ****	57.9	1.60; brs; 3H	17.9
10′	-	39.0	-	38.3	-	37.4	1.68; brd; 1.4; 3H *	25.9
11′a	4.24; dd; 5.5, 10.2; 1H ***	66.7	4.49; dd; 1.3, 10.2; 1H	67.0	4.39; dd; 4.9, 9.4; 1H	68.0	1.64; brs; 3H	16.2
11′b	4.22; dd; 6.6, 10.2; 1H ***		4.26; dd; 4.4, 10.2; 1H		4.26; t; 9.4; 1H			
12′a	4.91; brd; 1.6; 1H	108.0	1.24; s; 3H	30.8	1.27; s; 3H	24.9	-	-
12′b	4.56; brd; 1.6; 1H						-	-
13′	1.02; s; 3H	28.5	1.02; s; 3H	28.5	0.95; s; 3H	28.5	-	-
14′	0.82; s; 3H	15.6	0.84; s; 3H	15.5	0.92; s; 3H	17.3	-	-
15′	0.85; s; 3H	15.4	1.21; s; 3H	16.9	0.99; s; 3H	16.2	-	-
1″, 2″	-	-	-	-	-	167.9; 128.4	-	-
3″, 4″, 5″	-	-	-	-	6.05; qq; 1.4, 7.2; 1H: 1.89; quint; 1.4; 3H: 1.99; dq; 7.2, 1.4; 3H	137.8; 15.9; 20.8	-	-

*, **, *** and **** indicate overlapped or partially overlapped signals.

**Table 3 pharmaceuticals-17-01254-t003:** Cytotoxic activities of eleven sesquiterpene derivatives isolated from the roots of *F. caspica*.

Compounds	IC_50_ Values (µM) ^a^		
	COLO 205	MCF-7	K-562
Kayserin A (**1**)	61.31 ± 1.63	81.68 ± 1.47	60.20 ± 0.76
Kayserin B (**2**)	28.18 ± 0.82	229.17 ± 28.75	214.51 ± 10.33
8′-*epi*-Kayserin B angelate (**3**)	60.38 ± 0.90	33.69 ± 0.02	38.41 ± 0.44
3-*epi*-Ferulin D (**4**)	5.76 ± 0.54	11.04 ± 0.10	10.83 ± 0.21
Ferulin D (**5**)	34.23 ± 0.93	60.25 ± 3.60	24.01 ± 0.75
7-Desmethylferulin D (**6**)	61.70 ± 2.33	15.09 ± 0.26	4.93 ± 0.54
Colladonin (**7**)	10.28 ± 0.86	43.69 ± 2.39	22.82 ± 0.76
Isosamarcandin (**8**)	209.2 ± 11.85	81.57 ± 1.96	297.86 ± 6.72
Farnesyl Furanocoumarin Derivative (FFC) (**9**)	49.69 ± 2.31	19.01 ± 0.56	19.22 ± 1.24
NDHAP (**10**)	91.61 ± 1.57	43.06 ± 1.2	49.09 ± 0.70
HFDHAP (**11**)	3.96 ± 0.01	1.98 ± 0.64	5.25 ± 0.20
Cisplatin (Cis) *	111.87 ± 3.11	64.22 ± 5.25	8.10 ± 0.25
Doxorubicin (Dox) *	0.08 ± 0.00	2.83 ± 0.25	0.33 ± 0.02

^a^ 50% inhibitory concentration in COLO 205, MCF-7, and K-562 cells in the test assay. Values are expressed as mean ± SD for three separate experiments. * Positive control.

**Table 4 pharmaceuticals-17-01254-t004:** The docking scores and interactions of the compounds within the amino acids of the active site of Bcl-xL.

No	Compound	Docking Score	Hydrogen Bond Interaction	π-π Stacking Interaction
**1**	HFDHAP (**11**)	−7.395	ARG 139 (x2)	PHE 105
**2**	3-*epi*-Ferulin D (**4**)	−5.696	X	X
**3**	7-Desmethylferulin D (**6**)	−5.141	ARG 139	PHE 105
**4**	A1293102	−12.686	SER106, LEU 108, ASN 136, ARG 139	SER145, PHE 146

**Table 5 pharmaceuticals-17-01254-t005:** Predicted ADME properties of cytotoxic sesquiterpenes of *Ferula caspica* using the QikProp module of Schrödinger Software (release 2022-3).

**Compound**	**MW ^a^**	**LogPo/w ^b^**	**Log BB ^c^**	**PMDCK ^d^**	**HOA% ^e^**	**Rule of Five ^f^**
HFDHAP (**11**)	372.503	4.763	−1.897	145.175	100.00	0
7-Desmethylferulin D (**6**)	382.499	5.097	−0.834	647.221	100.00	1
3-*epi*-Ferulin D (**4**)	396.525	5.706	−0.257	2365.400	100.000	1
A1293102	972.116	6.078	−3.009	37.913	43.513	3

^a^: Molecular weight (recommended value: 150 to 500); ^b^: Octanol/water partition coefficient (recommended value: −2 to 5); ^c^: Brain–blood partition coefficient (recommended value: −3 to 1.2); ^d^: Permeability Maden–Darby canine kidney (<25 is poor, >500 is great); ^e^: HOA: Human oral absorption (≥80% is high, ≤25% is poor); ^f^: Number of violations of Lipinski’s rule of five.

**Table 6 pharmaceuticals-17-01254-t006:** Predicted toxicities of HFDHAP (**11**), 7-desmethylferulin D (**6**), and 3-*epi*-ferulin D (**4**) using ProTox-3.0.

Compound	SMILES	LD_50_ (mg/kg) (ProTox-3.0)
HFDHAP (**11**)	C\C(=C/CC(O)C(=O)c1ccc(O)cc1O)CC\C=C(/C)CC\C=C(/C)C	690
7-Desmethylferulin D (**6**)	C[C@@]1(OC=2Oc3cc(O)ccc3C(=O)C=2C1C)CC\C=C(/C)CC\C=C(/C)C	600
3-*epi*-Ferulin D (**4**)	COc1cc2OC=3O[C@](C)(CC\C=C(/C)CC\C=C(/C)C)C(C)C=3C(=O)c2cc1	600

## Data Availability

Data are contained within the article and Appendix A.

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
