# Peer review of "Sesquiterpene Coumarins, Chromones, and Acetophenone Derivatives with Selective Cytotoxicities from the Roots of Ferula caspica M. Bieb. (Apiaceae)"

_pharmaceuticals, 2024, doi:10.3390/ph17101254_

Round 1

Reviewer 1 Report

Comments and Suggestions for Authors

I have to admit that the manuscript "Sesquiterpene Coumarins, Chromones, and Acetophenone Derivatives with Selective Cytotoxicities from the Roots of Ferula caspica M. Bieb. (Apiaceae)" by Kuran et al. is one of the finest manuscripts I have reviewed for this journal. Everything here makes sense, from the very clear motivation to the proper choice of methods, a large number of techniques used to elucidate the structures of new compounds and test their biological activities, all the way to well-written conclusions. The findings, while not spectacular, definitely add to the existing knowledge and may be important for further research of compounds with high cytotoxicity.

I have just a few minor comments:

1. There are some small typos, like "RMSD 281 was determined to be 0.561" is missing the units (Angstroms); please reread the manuscript and correct all of them.

2. In figure 8 figures A1, B1, C1 are not legible and need to be of higher quality.

3. ADME part is fine, but I would try adding estimates of toxicity, even just by using the free ProTox 3.0 server - perhaps some interesting results can be added in this way.

I recommend a revision, but would really like to see this study published.

Comments on the Quality of English Language

Quality of English is fine.

Author Response

Comments and Suggestions for Authors

I have to admit that the manuscript "Sesquiterpene Coumarins, Chromones, and Acetophenone Derivatives with Selective Cytotoxicities from the Roots of Ferula caspica M. Bieb. (Apiaceae)" by Kuran et al. is one of the finest manuscripts I have reviewed for this journal. Everything here makes sense, from the very clear motivation to the proper choice of methods, a large number of techniques used to elucidate the structures of new compounds and test their biological activities, all the way to well-written conclusions. The findings, while not spectacular, definitely add to the existing knowledge and may be important for further research of compounds with high cytotoxicity.

I have just a few minor comments:

Dear Referee,

Thank you very much for your kind and constructive comments and suggestions.

  1. There are some small typos, like "RMSD 281 was determined to be 0.561" is missing the units (Angstroms); please reread the manuscript and correct all of them.

Response: The missing unit has been added to the revised manuscript.

  1. In figure 8 figures A1, B1, C1 are not legible and need to be of higher quality.

Response: The figure was revised to increase resolution. It is legible and included in the revised manuscript.

  1. ADME part is fine, but I would try adding estimates of toxicity, even just by using the free ProTox 3.0 server - perhaps some interesting results can be added in this way.

Response: Thank you for this constructive suggestion. We used ProTox 3.0 to assess the toxicities of the most potent sesquiterpenoid compounds; the results of this study are also included in the revised manuscript.

I recommend a revision, but would really like to see this study published.

Comments on the Quality of English Language

Quality of English is fine.

Submission Date

21 August 2024

Date of this review

30 Aug 2024 15:59:17

Reviewer 2 Report

Comments and Suggestions for Authors

This is the study that focused on identifying cytotoxic compounds from Ferula caspica with potential cancer treatment applications. However, there are some concerns that need to be addressed.

1. 1. While the mechanism of action is mentioned, it seems to rely heavily on molecular docking, the interaction between compound and Bcl-2 should be validated with more direct biochemical assays.

2.  2. The duration of treatment for each compound used to obtain the results in Table 1 needs to be specified. 

3.   3. To improve clarity, please include the compound codes in all figures. This will help readers easier to identify the different compounds being discussed.

4.    4. The WB results might not be fully convincing. The rationale behind the selected concentrations should be elaborated.

Author Response

Comments and Suggestions for Authors

This is the study that focused on identifying cytotoxic compounds from Ferula caspica with potential cancer treatment applications. However, there are some concerns that need to be addressed.

Dear Referee,

Thank you very much for your kind and constructive comments and suggestions.

  1. 1. While the mechanism of action is mentioned, it seems to rely heavily on molecular docking, the interaction between compound and Bcl-2 should be validated with more direct biochemical assays.

Response: Additional advanced biochemical analyses on Bcl-xL could not be conducted due to the limited quantity of isolated compounds from Ferula caspica and the need for more sufficient project funding. Future projects will be planned to validate the molecular docking and Western blot analysis results obtained from this study, as well as more advanced analyses and investigations of the compounds' effects on other proteins related to the apoptosis pathway.

  1. 2. The duration of treatment for each compound used to obtain the results in Table 1 needs to be specified. 

Response: The duration of the treatment was in Table 1 as footnote in the revised manuscript.

  1.  3. To improve clarity, please include the compound codes in all figures. This will help readers easier to identify the different compounds being discussed.

Response: The figures were revised to include compound codes in the revised manuscript.

  1.   4. The WB results might not be fully convincing. The rationale behind the selected concentrations should be elaborated.

Response: We understand the importance of transparency in our research methodology. The rationale behind the selected concentrations is as follows: Preliminary data from Western blot analyses of cisplatin indicated that a sufficient amount of cell lysate could not be obtained at concentrations above 130 µM. However, at 130 µM, an adequate amount of cell lysate was successfully obtained, and induction/suppression of the relevant proteins was observed. Consequently, 130 µM was established as the maximum dose. Given that substances 4, 6, and 11 are more toxic than cisplatin, dilution ratios of 1/5 and 1/50 (26 µM and 2.6 µM) were also chosen as working concentrations.

Submission Date

21 August 2024

Date of this review

30 Aug 2024 18:37:53

Reviewer 3 Report

Comments and Suggestions for Authors

The experimental work presented to me for evaluation concerns phytochemical and cytotoxicity studies of products isolated from a plant that occurs quite commonly in central Asia and reaches as far as the Mediterranean Sea. The genus Ferula has many species that have been studied more or less thoroughly. The authors studied the slightly less well-described species Ferula caspica. The article they prepared describes the isolation and study of 4 completely new chemical entities and the isolation of several other previously described compounds. The isolation process itself is quite complex and I am afraid that despite the detailed description it may be difficult to reproduce and detailed repetition. However, the final effects obtained and described by the authors are important for this work, and they do not raise any objections.

This work, from the scientific side, is prepared as correctly as possible, contains all the necessary elements in the right proportions and is almost perfectly developed and presented. Almost everything in this work meets the condition for its acceptance in its current form (but only in the scientific aspect !!). Unfortunately, the work was not free from certain significant carelessness and resulting errors. There are:

1. The numbering of the Figures themselves and the references to them in the text should be corrected very carefully. There are many errors in this respect.

2. On page 3, line 185 should be included in the "footnote".

3. The authors practically do not use subscript and superscript characters at all, writing the entire text, and especially chemical formulas and symbols, using characters of one type and size only.

4. It is a pity that the authors did not determine the melting point for the new substances. This is a parameter that is specific and, most importantly, unpredictable, characterizing a given chemical compound.

The above comments, although very important, practically concern only the technical preparation of the text. I have no reservations about the scientific side of the article. I believe that after this minor correction, the work can be accepted for publication without being re-evaluated by reviewers.

Author Response

Comments and Suggestions for Authors

The experimental work presented to me for evaluation concerns phytochemical and cytotoxicity studies of products isolated from a plant that occurs quite commonly in central Asia and reaches as far as the Mediterranean Sea. The genus Ferula has many species that have been studied more or less thoroughly. The authors studied the slightly less well-described species Ferula caspica. The article they prepared describes the isolation and study of 4 completely new chemical entities and the isolation of several other previously described compounds. The isolation process itself is quite complex and I am afraid that despite the detailed description it may be difficult to reproduce and detailed repetition. However, the final effects obtained and described by the authors are important for this work, and they do not raise any objections.

Dear Referee,

Thank you very much for your kind and constructive comments and suggestions.

This work, from the scientific side, is prepared as correctly as possible, contains all the necessary elements in the right proportions and is almost perfectly developed and presented. Almost everything in this work meets the condition for its acceptance in its current form (but only in the scientific aspect !!). Unfortunately, the work was not free from certain significant carelessness and resulting errors. There are:

  1. The numbering of the Figures themselves and the references to them in the text should be corrected very carefully. There are many errors in this respect.

Response: Thank you very much for pointing out the typographic mistakes; all of them have been corrected in the revised manuscript.

  1. On page 3, line 185 should be included in the "footnote".

Response: The footnote is added to Table 3.

  1. The authors practically do not use subscript and superscript characters at all, writing the entire text, and especially chemical formulas and symbols, using characters of one type and size only.

Response: The manuscript was revised to correct all superscript and subscript characters.

  1. It is a pity that the authors did not determine the melting point for the new substances. This is a parameter that is specific and, most importantly, unpredictable, characterizing a given chemical compound.

Response: Due to the extremely small quantities, our attempts to crystallize the novel compounds failed.

The above comments, although very important, practically concern only the technical preparation of the text. I have no reservations about the scientific side of the article. I believe that after this minor correction, the work can be accepted for publication without being re-evaluated by reviewers.

Submission Date

21 August 2024

Date of this review

29 Aug 2024 17:54:46

Round 2

Reviewer 1 Report

Comments and Suggestions for Authors

Authors have addressed all of my concerns so in my opinion the manuscript is ready to be published.

Reviewer 2 Report

Comments and Suggestions for Authors

The reviewer's concerns have been appropriately addressed, and there are no further comments.